# Research on Very-Low-Frequency Hydroacoustic Acoustic Velocity Sensor Based on DFB Fiber Laser

**Chenxia Ruan** [1,2,†], **Mo Chen** [3,†], **Yang Yu** [2], **Yichi Zhang** [3,4], **Jianfei Wang** [3,\*], **Zhenrong Zhang** [1,\*], **Junbo Yang** [5], **Shuolong Zhu** [1] and **Boyuan Qu** [1]

1   Guangxi Key Laboratory of Multimedia Communications and Network Technology, School of Computer, Electronic and Information, Guangxi University, Nanning 530004, China; zhushuolong@st.gxu.edu.cn (S.Z.); 2007210138@st.gxu.edu.cn (B.Q.)
2   College of Advanced Interdisciplinary Studies, National University of Defense Technology, Changsha 410073, China
3   College of Meteorology and Oceanography, National University of Defense Technology, Changsha 410073, China
4   Academy of Artillery and Air Defense, Nanjing 210000, China
5   College of Liberal Arts and Sciences, Notional University of Defense Technology, Changsha 410073, China
*   Correspondence: wangjianfei09@nudt.edu.cn (J.W.); zzr76@gxu.edu.cn (Z.Z.)
†   These authors contributed equally to this work.

**Abstract:** We have conducted a study on a very-low-frequency acoustic-velocity sensor which is based on a cantilever of distributed-feedback (DFB) fiber laser immersed in castor oil. A mathematical model of the frequency dependent response of the proposed sensor to the acoustic pressure signal influenced by the fluid viscosity is established. We have fabricated the proposed sensor and conducted experimental measurements in the standing wave tube. The results show that the sensor has an average phase sensitivity of $-179.5$ dB (0 dB = 1 rad/μPa) with ±1.45 dB fluctuation over the frequency range of 20–38 Hz. It has good cosine directivity with a directivity index of 32.5 dB and axial maximum asymmetry of 0.4 dB. The sensor presents promising applications for detecting very-low-frequency underwater acoustic signals.

**Keywords:** velocity sensor; DFB fiber laser; cantilever beam; phase sensitivity

## 1. Introduction

A fiber-optic vector sensor is one of the key research directions in the field of underwater acoustic technology, which is used for measuring hydroacoustic vector information, such as particle velocity, particle acceleration, particle displacement, and sound pressure gradient. The fiber-optic vector sensor is divided into two types of structures: the differential pressure type [1,2] and the co-vibration type [3–18]. Currently, the co-vibration fiber-optic vector sensor is the most commonly used type, which includes a displacement-based vector sensor [3–5], an acceleration-based vector sensor [6–13], and a velocity-based vector sensor [17,18]. However, the displacement-based vector sensor has a complex structure and low engineering measurement accuracy, making it unsuitable for practical applications. The acceleration-based vector sensor is the most common type of vector sensor. Unfortunately, its phase sensitivity decreases with decreasing frequency, which is not conducive to the detecting very-low-frequency underwater vector signals [15]. On the other hand, the velocity sensor has a flat phase sensitivity curve. However, the most reports on the velocity vector sensors were based on moving coil [19,20], piezoelectric [21], and magnetostrictive mechanisms [22]. Moving coil sensors were typically large in size and difficult to suspend, while poled polyvinylidene fluoride (PVF2) bimorphs designed for velocity hydrophones were small in size but had a sensitivity of only $-190$ dB (re 1 V/μPa). There have been few reports on fiber-optic velocity sensors. In 2012 and 2013, G. A. Cranch studied and reported the interaction between fluid-loaded fiber cantilevers and an acoustic

wave [17,18]. The results were very interesting, suggesting the potential for designing an acoustic velocity sensor.

In this paper, we investigate a very-low-frequency underwater acoustic velocity sensor that utilizes a cantilever structure consisting of a thin aluminum ribbon and a distributed feedback (DFB) fiber laser. We establish a mathematical model to calculate the frequency-dependent response of the proposed sensor to the acoustic pressure signal, which is influenced by the fluid viscosity. We then conduct experimental studies on the proposed sensor, packaged in a spherical shell with or without castor oil. The results demonstrate that the sensor without castor oil shows a flat frequency response to acoustic acceleration while the sensor with castor oil exhibits a flat acoustic response in the frequency range of 20–38 Hz. Furthermore, due to its characteristic of good cosine directivity, the fiber vector sensor is indicated to have a flat frequency response to the acoustic velocity.

## 2. Structure and Principle

### 2.1. Sensor Structure

The sensor system proposed in this study is illustrated in Figure 1. It consists of a thin aluminum ribbon and a DFB fiber laser that are combined to form the cantilever beam. The DFB fiber laser is attached to the center of the aluminum ribbon, and one end of the cantilever is fixed on the aluminum support while the other end is free. The DFB fiber laser is pumped by a 980 nm pump laser through a 980/1550 nm wavelength division multiplexer (WDM) to generate laser light. The laser light is then injected to an unbalanced Michelson interferometer through one port of the WDM. The interferometer is connected to a demodulation system for phase demodulation.

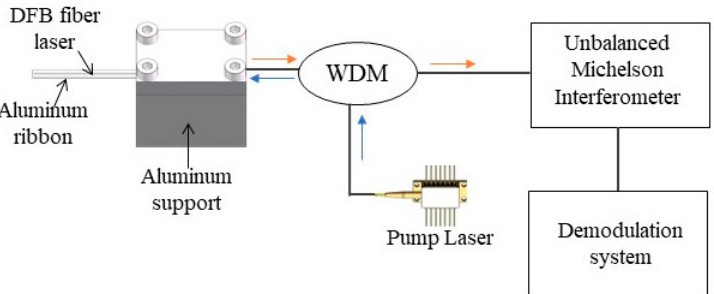

**Figure 1.** The schematic of the sensor system.

The vector sensor operates on the principle of cantilever deflection by fluid motion. The cantilever can be actuated via two methods, directly by the fluid motion when it is immersed in fluid or by applying a displacement to the support. In either case, the acoustic wave's driving force flexurally deforms the cantilever, causing the resonant cavity of the DFB fiber laser to also deform, resulting in a frequency variation of the fiber laser [23]. To measure this frequency variation, an unbalanced Michelson interferometer is used. The interferometer's output optical phase shift is given by [24]:

$$\Delta\phi = \frac{2\pi D \Delta v}{c} \tag{1}$$

where $D = 2nl$ is the optical path difference of the interferometer, $n$ is the refractive index of the fiber core, $l$ is the arm difference of the unbalanced Michelson interferometer, $\Delta v$ is the frequency variation of the fiber laser, and $c$ is the light speed in vacuum.

By measuring the phase of the interferometer (i.e., the frequency variation of the fiber laser), we can determine the driving force of the acoustic wave applied to the vector sensor. In the following section, we will delve into the detailed interactions between the fluid loaded fiber cantilevers and the acoustic wave.

### 2.2. Demodulation Algorithm

In this paper, we utilize the optimized $3 \times 3$ demodulation algorithm for demodulation. Compared with the traditional $3 \times 3$ coupler demodulation algorithm, the optimized $3 \times 3$ demodulation algorithm reduces 1 demodulation operation. Additionally, the optimized $3 \times 3$ demodulation algorithm greatly shortens the time required for sound field reduction demodulation. The orthogonal optimized $3 \times 3$ coupler interferometer demodulation method is shown in Figure 2.

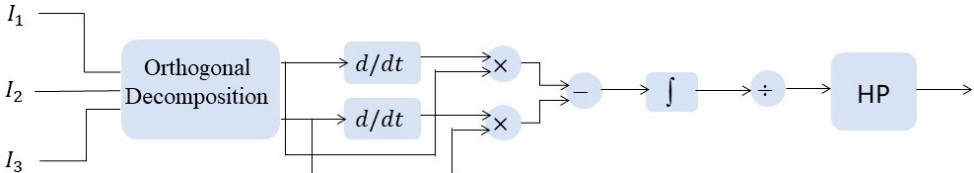

**Figure 2.** Interference demodulation method based on $3 \times 3$ coupler.

The light intensity detected by detectors $I_1$–$I_3$ can be expressed as,

$$
\begin{aligned}
I_1 &= \tfrac{2A_0^2}{9}[1 + \cos \varphi] \\
I_2 &= \tfrac{2A_0^2}{9}[1 + \cos(\varphi - \tfrac{2\pi}{3})] = \tfrac{2A_0^2}{9}[1 + \cos \varphi \cos \tfrac{2\pi}{3} + \sin \varphi \sin \tfrac{2\pi}{3}] \\
I_3 &= \tfrac{2A_0^2}{9}[1 + \cos(\varphi - \tfrac{4\pi}{3})] = \tfrac{2A_0^2}{9}[1 + \cos \varphi \cos \tfrac{4\pi}{3} + \sin \varphi \sin \tfrac{4\pi}{3}]
\end{aligned}
\tag{2}
$$

Equation (2) can be transformed into the matrix,

$$
\begin{bmatrix} I_1/I_4 \\ I_2/I_4 \\ I_3/I_4 \end{bmatrix} = \frac{2}{3} \begin{bmatrix} 1 & 0 & 1 \\ \cos \tfrac{2\pi}{3} & \sin \tfrac{2\pi}{3} & 1 \\ \cos \tfrac{4\pi}{3} & \sin \tfrac{4\pi}{3} & 1 \end{bmatrix} \begin{bmatrix} \cos \varphi(t) \\ \sin \varphi(t) \\ 1 \end{bmatrix}
\tag{3}
$$

where $I_4 = A_0^2/3$. The two orthogonal components $a = I_0 \cos \varphi(t)$, $b = I_0 \sin \varphi(t)$ are,

$$
\begin{bmatrix} a \\ b \\ 1 \end{bmatrix} = \begin{bmatrix} \cos \varphi(t) \\ \sin \varphi(t) \\ 1 \end{bmatrix} = \frac{2}{3} \begin{bmatrix} 1 & 0 & 1 \\ \cos \tfrac{2\pi}{3} & \sin \tfrac{2\pi}{3} & 1 \\ \cos \tfrac{4\pi}{3} & \sin \tfrac{4\pi}{3} & 1 \end{bmatrix}^{-1} \begin{bmatrix} I_1/I_4 \\ I_2/I_4 \\ I_3/I_4 \end{bmatrix}
\tag{4}
$$

After two identical differentiators, the following expressions are obtained:

$$
\begin{aligned}
c &= -I_0 \varphi'(t) \sin \varphi(t) \\
d &= I_0 \varphi'(t) \sin \varphi(t)
\end{aligned}
\tag{5}
$$

Then, multiply each way signal $a$ and $b$ with the other way differential and make the difference, which is given by:

$$
N = ad - bc = I_0^2 \varphi'(t)
\tag{6}
$$

In the actual environment, light source intensity fluctuations and polarization state changes will cause the value of $I_0$ to change, in order to eliminate the impact of $I_0$, the 2 input signals are squared to obtain:

$$
M = a^2 + b^2 = I_0^2
\tag{7}
$$

Then, divide $N$ by $M$ and eliminate $I_0^2$ to obtain:

$$
P = N/M = \varphi'(t)
\tag{8}
$$

After the integration operation, the output is as follows:

$$V_{out} = \sqrt{3}[\varphi(t) + \Psi(t)] \tag{9}$$

$\Psi(t)$ is the phase difference generated by the mathematical integration, and it is common to treat $\Psi(t)$ as a slow-varying quantity, which is filtered by a high-pass filter to demodulate the signal to be measured, $\varphi(t)$.

### 2.3. Theoretical Model of the Velocity Sensor

The geometry, cross-section, and displacement diagrams of the cantilever are shown in Figure 3. The cantilever's length, width, and height are denoted by $L$, $b$, and $h$, respectively. The DFB fiber laser adhered to the aluminum ribbon has a starting point $x_1$ and an ending point $x_2$. Its cross-section is a composite structure of a circle and a rectangle, shown in Figure 3b. The distance from the center of the fiber core to the neutral plane of the aluminum ribbon is denoted by $d_n$. Figure 3c shows $y_s(t)$ as the displacement of the support and $y(x,t)$ as the absolute displacement of the point $x$ on the cantilever at time $t$. We neglect all torsional effects in the cantilever and considers only the flexural modes of vibration, specifically those whose motion is strictly in the y-direction, as shown in Figure 3a.

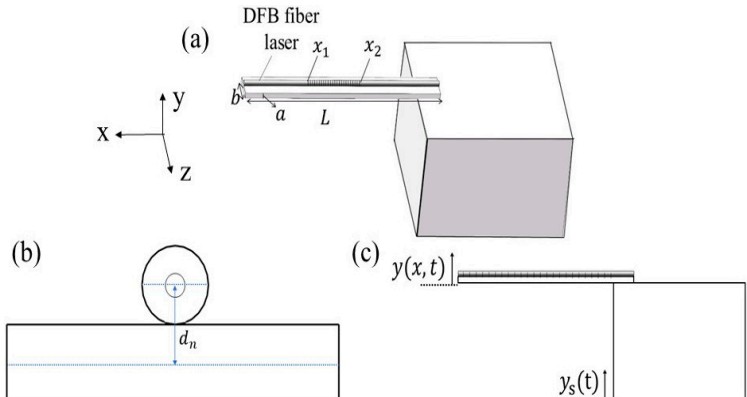

**Figure 3.** The structure of the cantilever. (**a**) Geometry diagram, (**b**) cross-sectional schematic, and (**c**) displacement diagrams.

The dynamic deflections of the cantilever, when the cantilever is immersed in viscous fluid and excited by acoustic waves, are modeled and analyzed by using the general theory given in Ref. [25]. The deflections of the cantilever are described by the beam equation:

$$EI\frac{\partial^4 y(x,t)}{\partial x^4} + m\frac{\partial^2 y(x,t)}{\partial t^2} = f(x,t) \tag{10}$$

$EI$ is the cantilever stiffness, $y(x,t)$ is the absolute displacement of the cantilever on the $y$-direction. $m$ is the mass per unit length of the cantilever. $f(x,t)$ is the external force on the cantilever. $t$ is the time variable. $x$ is the spatial variable that varies along the length of the cantilever.

There are four boundary conditions that apply to a cantilever structure with one end fixed: (1) the fixed end of the cantilever beam does not experience any deflection; (2) the derivative of the deflection at the fixed end is zero; (3) there is no bending moment at the free end of the cantilever, so the second-order derivative at the free end is zero; (4) there is no shearing force acting at the free end of the cantilever, so the third-order derivative is also zero. By applying these four boundary conditions, Equation (10) can be solved.

It is convenient to transform Equation (10) into the frequency domain to analyze the steady-state behavior of a cantilever in a viscous fluid. The motion form is assumed to be simple harmonic motion, $y(x,t) = y_0(x)\exp(i\omega t)$, then Equation (10) becomes:

$$EI\frac{\partial^4 Y(x|\omega)}{\partial x^4} - m\omega^2 Y(x|\omega) = F(x|\omega) \tag{11}$$

where $Y(x|\omega)$ is the Fourier transform of $y(x,t)$. $F(x|\omega)$ is the Fourier transform of $f(x,t)$, which consists of two parts:

$$F(x|\omega) = F_h(x|\omega) + F_d(x|\omega) \tag{12}$$

where $F_h(x|\omega)$ is the hydrodynamic load caused by the fluid motion around the cantilever, which is given by:

$$F_h(x|\omega) = m'\Gamma(\omega)\omega^2 Y(x|\omega) \tag{13}$$

$m'$ is the increased mass per unit length of the fluid, $m'\Gamma(\omega)$ is the 'virtual' mass of the fluid, where $m' = \pi\rho_{fl}b^2/4$, and $\Gamma(\omega)$ is the hydrodynamic function [25]. $\Gamma(\omega)$ is determined by Reynolds number Re. The Reynolds number Re of the acoustically induced fluid can be defined in terms of the main dimensions of the cantilever and is given by [25]:

$$\text{Re} = \frac{\rho_{fl}\omega b^2}{4\eta} \tag{14}$$

where $\rho_{fl}$ is the fluid viscosity, $\omega$ is the acoustic angular frequency, and $\eta$ is the dynamic viscosity of the fluid. The viscous effect has important implications for the operating frequency of the sensor due to the dependence of Re on $\omega$.

$F_d(x|\omega)$ is the driving force generated by the motion of the support. The cantilever is brought into motion by applying a displacement to the support, as shown in Figure 3c. The device measures the relative displacement between the support and the cantilever, which is given by $W(x|\omega) = Y(x|\omega) - Y_s(\omega)$. Therefore, the equation of motion in this case is given by:

$$EI\frac{\partial^4 W(x|\omega)}{\partial x^4} - (m + m'\Gamma(\omega))\omega^2 W(x|\omega) = F_d(x|\omega) \tag{15}$$

The driving force $F_d(x|\omega)$ generated by the movement of the support is expressed by the following formula.

$$F_d(x|\omega) = (m + m'\Gamma(\omega))\omega^2 Y_s(\omega) \tag{16}$$

$Y_s(\omega)$ is the displacement of the support. Theoretically, $m'\Gamma(\omega)$ is much larger than $m$. Thus, according to Equation (16), the response under pedestal drive is similar to that under fluid motion when $Y_s(\omega) = Y(x|\omega)$.

From the above analysis results, it can be seen that the plane acoustic wave generates a pressure gradient parallel to the motion of the cantilever, resulting in a driving force. The force per unit length of a small volume of fluid in the sound field is expressed as:

$$d\vec{f}(x,y,z) = -\nabla p(x,y,z)dA \tag{17}$$

where $p$ is the acoustic pressure, $dA$ is the cross-sectional area which the force acts, and $\nabla = \partial/\partial x \cdot \vec{i} + \partial/\partial y \cdot \vec{j} + \partial/\partial z \cdot \vec{k}$. According to the linear inviscid force equation for small-amplitude acoustic processes [26,27], the pressure gradient is also proportional to the fluid acceleration:

$$\rho_{fl}\frac{\partial \dot{u}_{fl}}{\partial t} = -\nabla p \tag{18}$$

where $\dot{u}_{fl}$ is the fluid velocity. A plane wave propagating in the $y$-direction has the following form, $p = p_0\exp(i(\omega t - ky))$, where $k$ is the acoustic wavenumber ($k = \omega/c_{fl}$) and $c_{fl}$ is the sound speed of the fluid. Assuming that the acoustic wavelength is much larger than the size of the cantilever, the Fourier transform of the force generated by the fluid motion is

represented by the acoustic pressure according to Equations (13) and (18), and the relation $df_y = -A_e \cdot \partial p / \partial y$.

$$F_{d-aco}(\omega) = -iA_e\Gamma(\omega) \bullet \frac{\omega}{c_{fl}}p_y \tag{19}$$

$A_e$ corresponds to the effective area of the calculated pressure gradient and is expressed by $A_e = m'/\rho_{fl}$.

The shape function $W(x|\omega)$ of the cantilever can be obtained by using the Green's function principle [25,28] and the boundary conditions calculation Equation (15), so that the strain $\Delta\varepsilon(x,t)$ generated by the core of the DFB fiber laser can be calculated. The equation is as follows:

$$\Delta\varepsilon(x,t) = d_n \bullet \frac{d^2 W(x|\omega)}{dx^2} \tag{20}$$

Among them, $d_n$ is the distance from the DFB fiber laser to the neutral axis of the aluminum ribbon, as shown in Figure 3b.

Thus, the optical frequency variation $\Delta v$ of the DFB fiber laser is obtained [29].

$$\frac{\Delta v}{v} = 0.78 \cdot q \int_{x_1}^{x_2} [\Delta\varepsilon(x,t) \times \exp(-2q|x - (x_1 + x_2)/2|)]dx \tag{21}$$

where $q$ is the coupling coefficient of the DFB fiber laser. $v$ is the laser frequency of the DFB fiber laser without bending strain. It is well known that directly measuring the optical frequency variation of the DFB fiber laser is a huge challenge. Therefore, to better measure the responses of the cantilever beam, the optical frequency variation is converted into the phase variation by using an interferometer. The phase variation $\Delta\varphi$ is given by Equation (1).

### 2.4. Simulations

To analyze the performance of the sensor, its sensitivities are simulated. The expressions of the acceleration sensitivity $S_a$ and the phase sensitivity $S_p$ are as following for the sensor:

$$S_a = 20\lg(\frac{\Delta\varphi}{a}) \tag{22}$$

where 0 dB = 1 rad/g. g is the gravitational acceleration of the earth, which is 9.81 m/s$^2$. $a$ is the acceleration change of the sensor movement.

$$S_p = 20\lg(\frac{\Delta\varphi}{p}) \tag{23}$$

where 0 dB = 1 rad/µPa. $p$ is the free-field acoustic pressure existing at the acoustic center of the sensor before the sensor is introduced into the sound field.

The frequency response of the sensor is calculated numerically. The DFB fiber laser is 30 mm long. It is adhered to an aluminum ribbon. The ribbon is 32 mm long, 3 mm wide, and 80µm thick. The unbalanced Michelson interferometer has an optical path difference of 1 m. The material parameters used in the calculations are presented in Table 1.

Initially, the sensor is assumed to operate in air with the cantilever being actuated by the support's motion. As air has a low viscosity, no fluid is loaded onto the cantilever. Figure 4 shows the theoretical results of the acceleration sensitivity of the sensor, where the resonate peak occurs at 126 Hz and the curve is flat between 5 and 80 Hz. The average acceleration sensitivity of the sensor is 27 dB (0 dB = 1 rad/g). These results indicate that the sensor behaves as a typical accelerometer in air without appropriate fluid loading on the cantilever.

**Table 1.** Definition of terms.

| Parameter | Definition | Value |
|---|---|---|
| Fluid properties | | |
| $\rho_{fl}$ | Density | 997 kg/m$^3$ (Castor oil)<br>1.29 kg/m$^3$ (Air) |
| $c_{fl}$ | Sound speed | 340 m/s (Air)<br>1540 m/s (Castor oil) |
| $\eta$ | Fluid viscosity | $1.8 \times 10^{-5}$ Pa·s (Air)<br>0.985 Pa·s (Castor oil) |
| Fiber properties | | |
| $\rho_f$ | Density | 2200 kg/m$^3$ |
| $E$ | Youngs modulus | 70 GPa |
| $r_f$ | Coating radius | $62.5 \times 10^{-6}$ m |
| $q$ | Grating coupling coefficient | 180 m$^{-1}$ |
| $c$ | Speed of light | $3 \times 10^8$ m/s |
| $\lambda$ | Wavelength | 1550 nm |
| $n$ | Core index of refraction | 1.47 |
| Ribbon properties | | |
| $E_{lv}$ | Youngs modulus | 70 GPa |
| $\sigma$ | Poisson's ratio | 0.33 |
| $\rho_{lv}$ | Density | 2700 kg/m$^3$ |
| $L$ | Length | 32 mm |
| $b$ | Width | 3 mm |
| $h$ | Thickness | 80 µm |

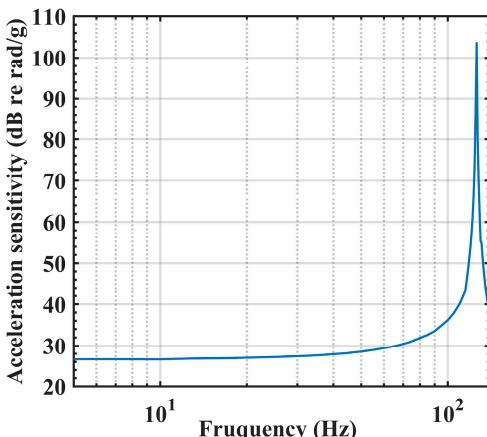

**Figure 4.** The acceleration sensitivity of the sensor.

Then, we consider the cases when the cantilever operates in different fluids with various viscosities, namely water, castor oil, and glycerin, as shown in Table 2. The simulated results of the phase sensitivity of the sensor are shown in Figure 5. For comparison, we also show the result when the cantilever operates in air in the figure. Since the viscous damping of the air is very small, it can be approximated as an inviscid liquid. The blue curve in Figure 5 represents the simulation when the sensor is in an inviscid fluid. The phase sensitivity increases with the frequency before the resonance peak appears. This is because the sensor without viscous fluid loading is an acceleration sensor and its phase sensitivity decreases with the decreasing frequency. When the sensor is immersed in different viscous fluids, the sensitivity exhibits different characteristics. As seen in Figure 5, the viscous fluids have a significant effect on the frequency response of the sensor. When the fluid is water, the sensitivity increases by about 10 dB compared to when the sensor operates in air. When the fluid is castor oil, the resonate peak of the response curve disappears, and the sensitivity of the sensor is flat in the frequency range of 20–80 Hz. This indicates that the sensor becomes a velocity sensor with castor oil loading on the cantilever. As

the fluid viscosity increases, the Reynolds number decreases and falls below 1, and the driving force becomes dominated by the imaginary part of the hydrodynamic function, $\Gamma(\omega)$. This leads to an increase in response with decreasing frequency and the cantilever is driven by the viscosity of the fluid instead of the fluid acceleration. If the viscosity is further increased, for example, if the fluid is glycerin with a viscosity of more than 1 Pa·s, the sensitivity of the sensor increases, but the frequency response is not very flat. A trough appears at 28 Hz. This is because the fluid viscous force exceeds the load capacity of the cantilever and affects the operating frequency range of the sensor. The sensor achieves a flat response in the frequency range of 20–80 Hz when the viscosity is 0.985 Pa·s (castor oil). Therefore, the sensor is expected to be packaged in a shell filled with castor oil for acoustic velocity sensing.

**Table 2.** Properties of viscous fluids.

| Liquid | Viscosity (Pa·s) | Density (Kg/m$^3$) |
|---|---|---|
| air | $1.8 \times 10^{-5}$ | 1.4 |
| water | $8.9 \times 10^{-4}$ | 998 |
| castor oil | 0.985 | 961 |
| glycerin | 1.5 | 1261 |

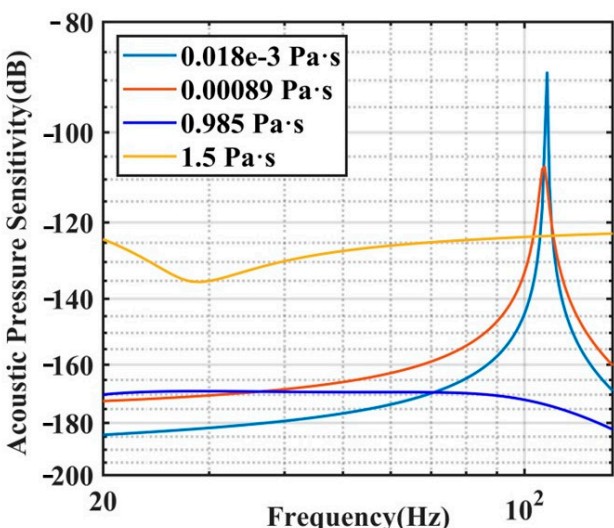

**Figure 5.** The phase sensitivity of the sensor.

## 3. Experimental Results and Discussions

### 3.1. Response of the Cantilever

We conducted experiments to verify the theoretical analysis. We fabricated a DFB fiber laser cantilever and tested its acceleration sensitivity in air. The cantilever is stably laid on the surface of a shaker for measurement of acceleration sensitivity, as shown in Figure 6.

The experimental setup for measuring the acceleration sensitivity of an unpackaged cantilever is shown in Figure 7. The cantilever is 32 mm long, and the aluminum ribbon is 3 mm × 80 μm in size. It is worth noting that the parameters of the aluminum ribbon affect the sensitivity and resonance frequency of the sensor [17]. The longer the cantilever length, the larger the sensitivity of the sensor, but the lower the resonance frequency. There is a trade-off between sensitivity and resonant frequency. Considering the resonant cavity of the adopted DFB fiber laser, which is 30 mm long, the length of the aluminum ribbon is chosen to be 32 mm. As for the width and thickness of the ribbon, we referred to Ref. [17]. The length, width, and height of the aluminum support are 3 cm, 2 cm, and 2 cm, respectively. The aluminum support is laid on the surface of the shaker. The shaker is driven by a sinusoidal voltage generated by an arbitrary signal generator and amplified by a power

amplifier to generate a vibration signal for the sensor. The vibration acceleration can be obtained by the standard accelerometer equipped in the shaker. The acceleration response of the cantilever can be measured when the driving force is from the motion of the support. The DFB fiber laser is pumped with a 980 nm pump laser and emits single-frequency light at around 1550 nm. The 1550 nm laser is output through the 1550 nm port of the WDM. A circulator is placed at the laser output to eliminate unwanted reflections on the DFB fiber laser. The laser enters an unbalanced 3 × 3 fiber-optic Michelson interferometer from port 2 of the circulator. The optical path difference of the unbalanced Michelson interferometer is 1 m. The frequency shifts of the fiber laser are transformed into the phase signal by the Michelson interferometer. The relationship between the frequency shifts and the phase shifts is given by Equation (1). The optical signal of the interferometer is received by the photodetector and sampled by the data acquisition card. The acceleration signal of the shaker received by the sensor is demodulation by software on the personal computer (PC). Additionally, the interferometer is placed in a shielding can to reduce the influence of environmental noise on the interferometer.

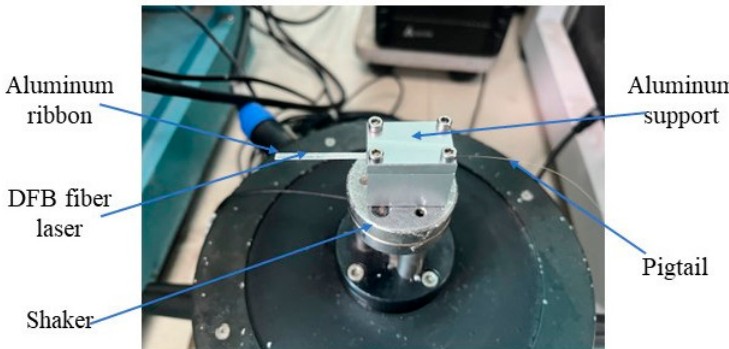

**Figure 6.** The setup of the aluminum support on the shaker.

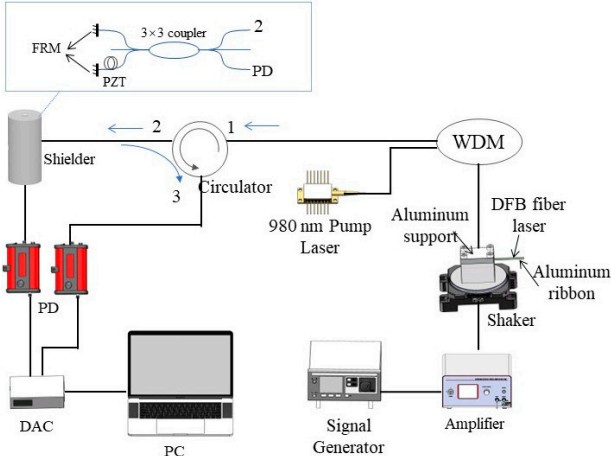

**Figure 7.** The experimental setup for acceleration sensitivity of the sensor.

The acceleration sensitivity of the planar cantilever was measured in the range of 5–140 Hz. The experimental results are shown by the red curve in Figure 8. The response is flat, and the average acceleration sensitivity is 27 dB (0 dB = 1 rad/g) between 5 and 80 Hz, which is consistent with the simulation. The extremely small viscous damping leads to no fluid load on the cantilever, indicating that the sensor has high acceleration sensitivity. A resonance peak is observed at 126 Hz, and the acceleration sensitivity deviates from the simulation. This deviation is due to the influence of the viscous damping of the air, which was not considered in the theoretical analysis. However, in the experiment, there is both structural damping and aerial damping in the composite structure. The vibration of

the support can distort the response of the cantilever near the 126 Hz frequency, and the structural damping of the adhesive coating or the fixture can also have an effect. Therefore, the measured resonance amplitude deviates from the theoretical value.

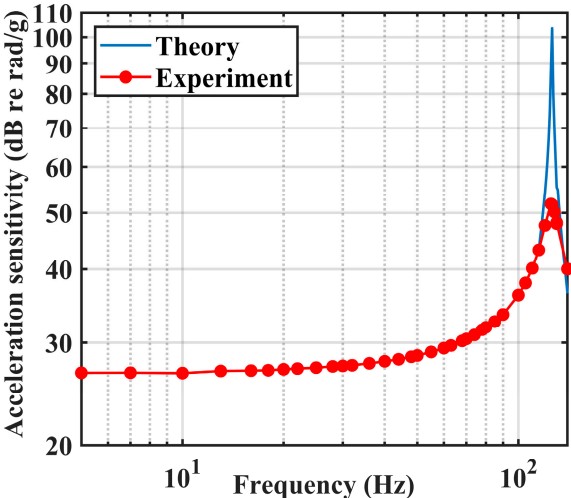

**Figure 8.** Acceleration sensitivity curve of the planar cantilever in air.

### 3.2. Performance of the Sensor

According to the theoretical results in Section 2.3, the cantilever should be immersed in castor oil to operate in the velocity sensor mode. Therefore, we packaged the cantilever in a sphere filled with castor oil. The cantilever support is rigidly attached to the inner wall of the sphere, as shown in Figure 9a. The packaged velocity sensor's appearance is shown in Figure 9b. The total density of the packaged sensor is nearly equal to that of water, making the sphere neutrally buoyant. When the sphere is driven by the acoustic wave in the water, it moves along with the water. The support of the cantilever is fixed rigidly to the sphere. The support is driven by the force of the acoustic wave. The sphere is filled with castor oil so that the cantilever is immerged in the castor oil. The motion of the cantilever satisfies with Equations (15) and (16).

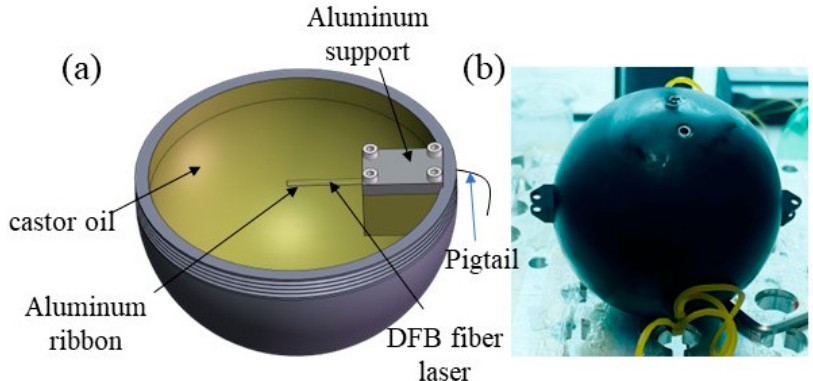

**Figure 9.** (**a**) The schematic of the velocity sensor. (**b**) Photograph of the assembled velocity sensor.

It is worth noting that the shell does not necessarily have to be spherical; it can be cylindrical or any other shape that does not significantly affect the cantilever's motion. The essential factor is that the sensor's density should be close to that of water, so it follows the motion of the water particle when suspended in water. The sphere's material is aluminum. The sensor is suspended in a standing wave tube for sensitivity measurement using some rubber bands on a vertical slewing device, as shown in Figure 10. There is a shaker on the bottom of the standing wave tube. The height from the water surface to the tube bottom is *l*. The shaker generates vibrations (i.e., sound) in the water. Because of the size of the tube

(tube diameter is larger than the height of the water in the tube), the sound in the tube is a standing wave. The pressure under the water surface is $P(x) = Ae^{jwt+kx} + Be^{jwt-kx}$. where $k$ is wave number, $A$ and $B$ is coefficients determined by the tube, $x$ is the distance from the water surface. When $x = 0$, $p(0) = 0$. When $x = l$, $P(l) = P_l e^{j\omega t}$. So the pressure at the depth of $d$ is $P_d = \frac{P_l \sin kd}{\sin kl} e^{j\omega t}$. The vector sensor and the standard hydrophone are at the same depth below the water surface, so they experience the same acoustic pressure. The data obtained in the measurement are validated. When a standard hydrophone is placed at the same depth as the vector sensor, the phase sensitivity of the vector sensor can be acquired by [30]

$$M_p = S_0 + 20\lg(\frac{\Delta\varphi}{U_0}\tan kd) \tag{24}$$

where $M_0$ is the sound pressure sensitivity of the standard hydrophone in dB, $\Delta\varphi$ the amplitude of the phase signal received by the vector sensor, $U_0$ is the amplitude of the voltage from the standard hydrophone, $k$ is the wave number, $d$ is the unified depth of the two hydrophones.

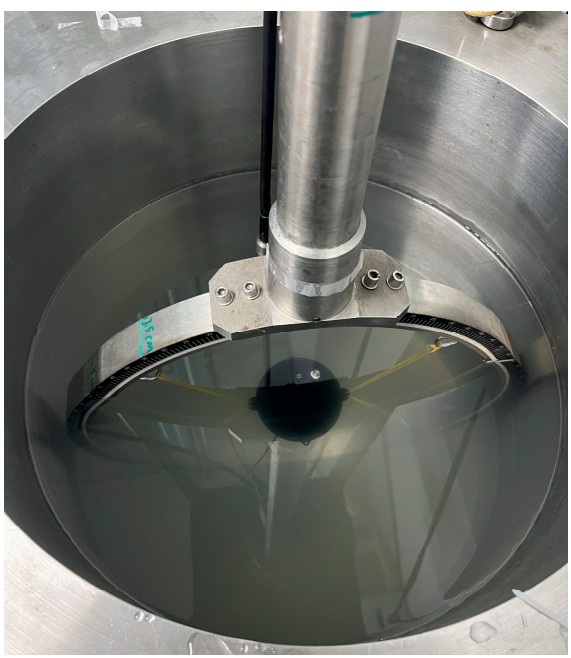

**Figure 10.** The practicality picture of the sensor in the standing wave tube.

The experimental setup for measuring the phase sensitivity of the sensor is shown in Figure 11. According to Equation (24), we put the vector sensor and the standard hydrophone (with the sensitivity of −190.8 dB re rad/μPa) at the same depth $d = 12.5$ cm. A sinusoidal voltage generated by an arbitrary signal generator and amplified by a power amplifier is utilized to drive the shaker at the bottom of the standing wave tube, generating an acoustic signal for the sensor.

The experimental results without castor oil in the sphere are shown in the blue curve of Figure 12a. The measured phase sensitivity is approximately −190.6 dB (0 dB = 1 rad/μPa) at 20 Hz. However, there is a resonance peak at 100 Hz, which is 26 Hz lower than the unpackaged. This may be attributed to the sound–structure coupling effect. When the sphere is excited, the aluminum support vibrates, generating a sound wave that acts on the cantilever. The cantilever amplifies or attenuates the sound wave, which subsequently reacts to the support. Consequently, the inherent mechanical vibration coupling of the support affects the DFB fiber laser, resulting in a low resonance frequency for the packaged sensor compared to the unpackaged sensor.

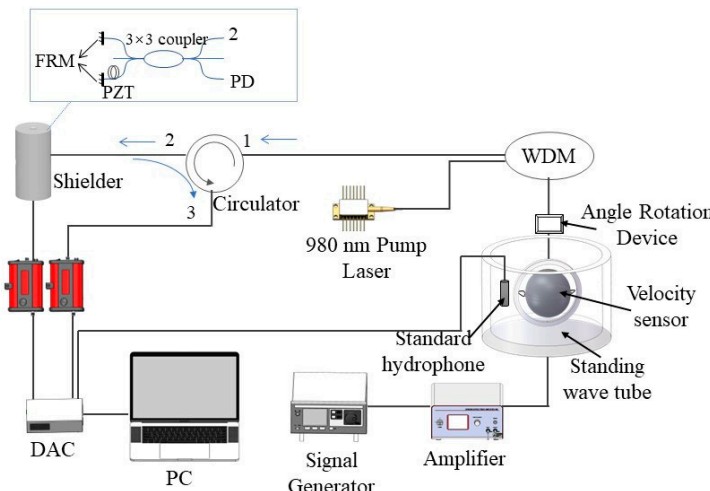

**Figure 11.** The experimental setup for phase sensitivity of the sensor.

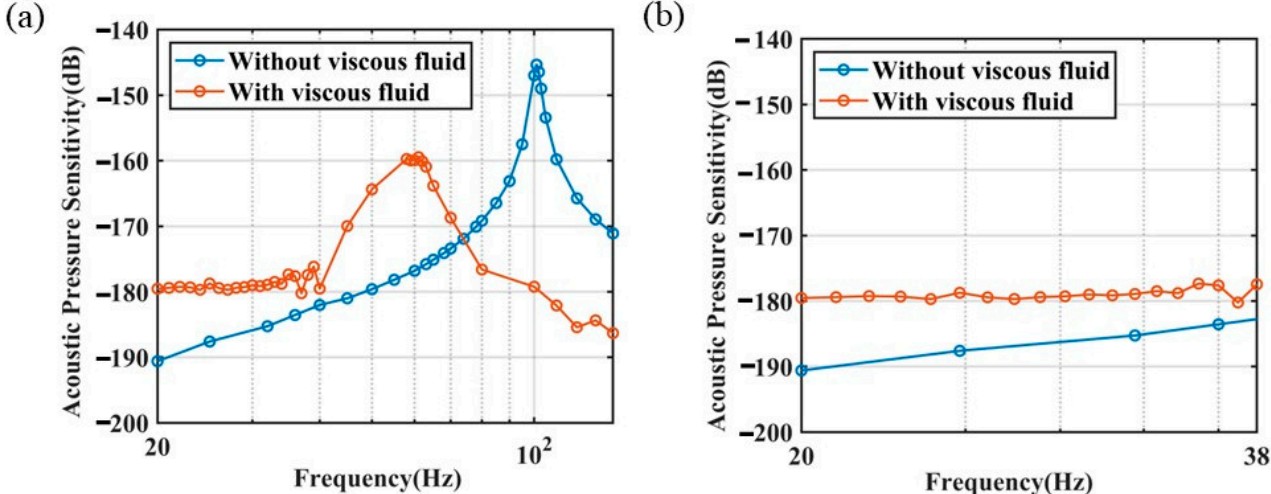

**Figure 12.** The phase sensitivity curve of the sensor. (**a**) 20–140 Hz; (**b**) 20–38 Hz.

The red curve in Figure 12a displays the phase sensitivity measurements when the sphere is filled with castor oil. The frequency response of the sensor is flat with a fluctuation of ±1.45 dB over the 20 to 38 Hz range. The phase sensitivity is −179.5 dB (0 dB = 1 rad/μPa) at 20 Hz, which is 11.1 dB higher than the measurement without castor oil, as illustrated in Figure 12b. The sensor changes from acceleration sensitive to velocity sensitive due to the damping effect of the castor oil altering the force on the cantilever. It is worth noting that the theoretical predictions in Figure 5 do not show a resonance peak around 60 Hz for the velocity sensor in Figure 12a. There is an apparent difference between the experimental results and the theoretical predictions. This could be due to the fabrication of the sensor, where the support of the cantilever is attached to the sphere's inner wall causing a mismatch between the barycenter and the center of mass for the velocity sensor. This mismatch may produce the fundamental resonance at 60 Hz. Further theoretical and experimental investigations are required to determine the reason, and future work will focus on upgrading and optimizing the structure and parameters for the velocity sensor.

The background phase noise is one of the crucial performance parameters of the fiber-optic vector sensor. The phase noise level is measured and presented in Figure 13, where the phase noise level is −64.8 dB (re rad/Hz$^{1/2}$) at a frequency of 32 Hz.

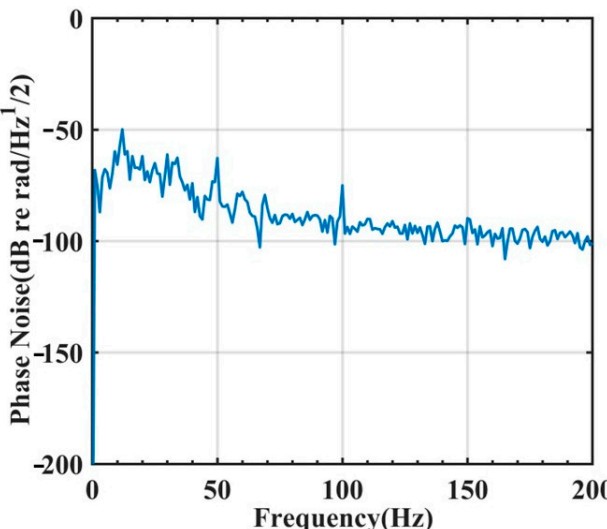

**Figure 13.** System background phase noise spectrum level.

Directivity is an important performance index for fiber-optic vector sensors, as it describes the sensitivity of the sensor to sound waves coming from different directions. The directional pattern of the sensor can be obtained by rotating the sensor around its center and measuring the magnitude of its output signal at different angles. In Figure 14, the directional pattern of the sensor is shown for a frequency of 32 Hz. The directivity index of the sensor is 32.5 dB and the maximum asymmetric index is 0.4 dB. This indicates that the underwater acoustic velocity sensor is only sensitive to the motion perpendicular to the cantilever plane.

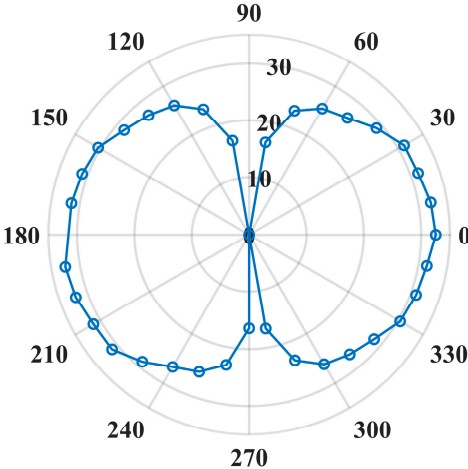

**Figure 14.** Directivity of the sensor measured at 32 Hz.

## 4. Conclusions

We have conducted a study on a fiber-optic underwater acoustic velocity sensor operating at very low frequencies. The phase sensitivity of the sensor in difference viscous fluids are analyzed theoretically, and the results indicated that the sensor responds to acoustic velocity when the cantilever is immersed in castor oil. We fabricated the sensor and measured its phase sensitivity in the standing wave tube. The experimental results showed that the average phase sensitivity of the sensor reached −179.5 dB (0 dB = 1 rad/μPa) with ±1.45 dB fluctuation in the frequency range of 20–38 Hz. The maximum axial asymmetry was 0.4 dB, and its directivity index was 32.5 dB. The background phase noise was −64.8 dB (re rad/Hz$^{1/2}$) at 32 Hz frequency. The experimental results indicated that the

vector sensor presented a flat frequency response to acoustic pressure, indicating its flat response to acoustic velocity. The fiber-optic velocity sensor has promising applications in very-low-frequency underwater acoustic signal detections.

**Author Contributions:** Conceptualization, C.R. and M.C.; methodology, J.W.; software, Y.Z.; validation, C.R., Y.Y. and M.C.; formal analysis, C.R. and J.W.; investigation, C.R. and S.Z.; resources, M.C., Y.Y., J.Y. and J.W.; data curation, C.R.; writing—original draft preparation, C.R.; writing—review and editing, M.C. and J.W.; visualization, C.R. and B.Q.; supervision, Z.Z.; project administration, Z.Z.; funding acquisition, Z.Z. All authors have read and agreed to the published version of the manuscript.

**Funding:** This research was funded by National Natural Science Foundation of China (Grant Nos. 62275269); Guangdong Guangxi Joint Science Key Foundation (2021GXNSFDA076001); Guangxi Major Projects of Science and Technology (grant No. 2020AA21077007 and 2020AA24002AA). This work supported by the interdisciplinary scientific research foundation of Guangxi university (Grant No. 2022JCC014).

**Institutional Review Board Statement:** Not applicable.

**Informed Consent Statement:** Not applicable.

**Data Availability Statement:** Data underlying the results presented in this paper are not publicly available at this time but may be obtained from the authors upon reasonable request.

**Acknowledgments:** The authors would like to thank the laboratory and university for their support.

**Conflicts of Interest:** The authors declare no conflict of interest.

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
