# Peer review of "Research on Very-Low-Frequency Hydroacoustic Acoustic Velocity Sensor Based on DFB Fiber Laser"

_photonics, doi:10.3390/photonics10040463_

Round 1

Reviewer 1 Report

In this paper, the authors have presented a distributed feedback (DFB) fiber laser-based acoustic-velocity sensor. I believe that the results are interesting. The manuscript can be considered further with revisions taking into account my following remarks.

1. Since the DFB fiber laser is an important device in this work, the authors should provide more details on DFB fiber laser. (e.g., principle, criteria, ect.)

2. The authors should describe in detail how the shaker obtains the signal in Figure 6. Then how to extract the vibration signal?

3. Figure 7: Apart from the peak sensitivity, the experimental and theoretical results are an excellent match. But why the effect of the adhesive have almost no effect at lower frequencies?

4. What are the criteria for selecting those viscous fluids? How are they connected to the phase sensitivity of the optical scheme connecting the DFB fiber laser and interferometer?

Author Response

Comments and Suggestions for Authors

In this paper, the authors have presented a distributed feedback (DFB) fiber laser-based acoustic-velocity sensor. I believe that the results are interesting. The manuscript can be considered further with revisions taking into account my following remarks.

1. Since the DFB fiber laser is an important device in this work, the authors should provide more details on DFB fiber laser. (e.g., principle, criteria, ect.)

Answer: Thanks very much for the reviewer’s suggestion. The DFB fiber laser is important for the sensor. But it is actually very mature devices. We do not think it is necessary to include the detailed principle of them, which will make the manuscript too long. Reference [23] is a classic literature on the DFB fiber laser. The classic literature in section 2.1 in case the readers wanted to learn deeply about the DFB fiber laser.

2. The authors should describe in detail how the shaker obtains the signal in Figure 6. Then how to extract the vibration signal?

Answer: Thanks to the reviewer’s question. The sinusoidal voltage generated by an arbitrary signal generator and amplified by a power amplifier, is used to drive the shaker to generate a vibration signal for the sensor. We have added the relevant contents in section 3.1.

The vibration in the vertical direction of the cantilever causes bending deformation of the cantilever, which in turn leads to strain in the resonant cavity of the DFB fiber laser. The strain in the resonant cavity leads to a change in the longitudinal mode in the fiber laser (i.e., a change in the output optical frequency). Therefore, the output optical frequency is modulated by the vibration signal, which can be demodulated by detecting the laser output optical frequency.

3. Figure 7: Apart from the peak sensitivity, the experimental and theoretical results are an excellent match. But why the effect of the adhesive has almost no effect at lower frequencies?

Answer: This is a very good question. The experimental results agree well with the theoretical results in Figure 8. The sensor is an accelerometer in Figure 8. It is normal for an accelerometer to realized flat acceleration sensitivity before the resonance peak even at lower frequency. It is normal. The fiber aluminum mismatch and the effect of the adhesive are tiny and not that obvious to affect the acceleration response.

4. What are the criteria for selecting those viscous fluids? How are they connected to the phase sensitivity of the optical scheme connecting the DFB fiber laser and interferometer?

Answer: This is a good question. Air, water, castor oil and glycerin are very typical and pretty common. Their viscosities varies from 1.8×10-8 to 1.5 Pa·s, nearly covering the fluid with no viscosity and high viscosity. The viscosity of air is very small, and there is almost no load on the cantilever. However, the viscosity of glycerin is 1.5 Pa·s, which makes the cantilever have a large bending. The effect of fluid viscosity on the sensing performance of the sensor can be well and overall studied by using these four different fluids.

In the presence of the viscous fluid surrounding the cantilever, the sensor is forced by the acoustic pressure force and an additional viscous drag force. The forces bend the cantilever and cause the frequency shift of the DFB fiber laser. The light of the fiber laser is injected to the interferometer and generates phase shift of the interferometer. The phase shift of the interferometer is measured and demodulated. The acoustic pressure signal can be measured by a standard pressure hydrophone, as shown in Figure 11. And then the phase sensitivity of the sensor can be obtained from Eq. (23).

Reviewer 2 Report

1.What is the optical path difference of the interferometer used in the simulation?

2.Briefly explain the demodulation method and correspond to it in the figure. The detectors in Figure 6 and Figure 10 are not consistent.

3.The background noise of the system is relatively high. What are the methods to reduce it in the follow-up study.

Author Response

Comments and Suggestions for Authors

1.What is the optical path difference of the interferometer used in the simulation?

Answer: Thanks to the reviewer’s question. In the manuscript, we should introduce the optical difference of the interferometer in detail in the simulation. The optical path difference of the interferometer is 1 m in the simulation. We also added relevant content in the second paragraph of section 2.4.

2.Briefly explain the demodulation method and correspond to it in the figure. The detectors in Figure 6 and Figure 10 are not consistent.

Answer: Thank you for the reviewers’ comments. We use the optimized 3×3 demodulation algorithm for demodulation. Compared with the traditional 3×3 coupler demodulation algorithm, the optimized 3×3 demodulation algorithm can be reduced 1 demodulation operation. In addition, the optimized 3×3 demodulation algorithm can be greatly shortened the time required for sound field reduction demodulation. The orthogonal optimized 3×3 coupler interferometer demodulation method is shown in Figure 1.

Figure 1. Interference demodulation method based on 3×3 coupler

The light intensity detected by detectors I1-I3 can be expressed as,

(1)

Eq. (1) can be transformed into the matrix, which is given by,

(2)

Where . The two orthogonal components ,  are,

(3)

After two identical differentiators, the following expressions are obtained,

(4)

Then multiply each way signal  and  with the other way differential and make the difference,which is given by,

(5)

In the actual environment, light source intensity fluctuations and polarization state changes will cause the value of  to change, in order to eliminate the impact of , the 2 input signals are squared to obtain,

(6)

Then divide  by  and eliminate  to obtain,

(7)

After the integration operation, the output is as follow,

(8)

 is the phase difference generated by the mathematical integration, and it is common to treat  as a slow-varying quantity,which is filtered by a high-pass filter to demodulate the signal to be measured, . We have added the relevant contents in section 2.2.

As to Figure 7 and Figure 11, we have modified the detector in Figure 11.

3.The background noise of the system is relatively high. What are the methods to reduce it in the follow-up study.

Answer: This is a good question. It is of importance to reduce the background noise of the system. We will use the pump source and fiber laser with lower noise level, which should be main way to reduce noise. The signal detection system can be optimized and detector with lower noise can be used. In addition, the package structure to minimize the bending of the fiber.

Reviewer 3 Report

This paper is on a low frequency fiber laser acoustic vector sensor based on a cantilever beam.

In the theoretical part interaction of the beam with acoustic fluid motion is considered in the presence of the beam support displacement. Modelling of the system has been performed for different types of fluid.

In experiment, the sensor response has been measured both in the air and in liquids.

Once the subject of the research is of an interest and the paper includes some detailed explanations as for the theoretical section and the experiment, there are some serious flaws in the paper. In my view, the paper can’t be published in its current form and needs major revision.

Here are some major points explaining this:

1.       There are some flaws with English. (lines 29-33, 56-57 or 274-275 etc). (Of course, a sphere filled with castor oil and fulfilled with castor oil is not the same thing. But I will not make emphasis just on English, leaving this to editors.

2.       On lines 19-22 (and in the rest of the paper) authors mention pressure sensitivity of the acoustic vector sensor, or acceleration sensor, etc. Ideally, the velocity sensor should not respond to pressure, it should be sensitive only to a single parameter, so this influence is purely parasitical (in this particular case, it may be the hydrostatic sensitivity of the fibre laser). But in the paper, authors call acceleration response or velocity response a “pressure sensitivity” for unknown reasons.

3.       Also, throughout the entire paper, and on all their graphs, they call sensor “sensitivity” what is in fact the sensor “response”. The sensitivity is the ability to sense smallest signals, and in the current case is limited to the laser noise. The graphs (all of them) also do not show units on vertical axis.

4.       The section 2.2 (theoretical model) seems to have very little connection with the rest of the text. For example, ln 157-160 authors claim that the shape of W(x/w) can be obtained but they do not present this, even for comparison with a well-known from mechanics formula of bent cantilever beam.

5.       In section 2.3 ( ln. 171-174) authors claim that formulas (14) (15) can be obtained from section 2.2. I believe it is wrong, and there is no such connection. Sa and Sp simply would follow from authors’ measurement setup (which was not though explained in enough detail)

6.       See ln.175 g- is not a “gravitational constant” (and never was), it is a free-fall acceleration near the Earth surface. And certainly, has nothing to do with the subject of this paper. I do not know why authors have chosen 9.8m/sec2 as a reference and not just 1m/sec2.

7.       There are serious flaws with the experiment (ln 295 onwards). The authors used a standard hydrophone as a reference for a vibration sensor, which is totally crazy, as they respond to different parameters. Moreover, the “standard hydrophone” is placed within 25cm of the water surface (pressure release surface) where (in theory), the acoustic pressure is zero. In current configuration, the bottom and the sides of their “standing wave tube) are the sources of secondary acoustic waves (in the sensors’ proximity) and it is not easy to predict the measurement result. In my view this invalidates any data obtained in the measurement.

8.       Statements in Ln 339-344 are plain wrong. You can’t enhance the sensor response (let alone the sensitivity) just by manipulating your demodulation system.

9.       Discussion on the fiber laser noise (ln345-347) seem to be totally out of context. But it is what actually defines the real “sensitivity” of the fiber laser sensor. I also totally do not understand why authors are always using “rad” units instead of “Hz” units throughout the entire paper. The phase is what is obtained by passing the fiber path imbalance, so why the sensor itself and its response (Hz/Pa, or Hz/(m/sec2)) is assumed to have connection with authors’ particular demodulation system (and the length of the path imbalance) is unclear.

Overall, given there are so many serious points (including the results of the core experiment), the paper requires serious revision

Author Response

Comments and Suggestions for Authors

This paper is on a low frequency fiber laser acoustic vector sensor based on a cantilever beam.

In the theoretical part interaction of the beam with acoustic fluid motion is considered in the presence of the beam support displacement. Modelling of the system has been performed for different types of fluid.

In experiment, the sensor response has been measured both in the air and in liquids.

Once the subject of the research is of an interest and the paper includes some detailed explanations as for the theoretical section and the experiment, there are some serious flaws in the paper. In my view, the paper can’t be published in its current form and needs major revision.

Here are some major points explaining this:

  1. There are some flaws with English. (lines 29-33, 56-57 or 274-275 etc). (Of course, a sphere filled with castor oil and fulfilled with castor oil is not the same thing. But I will not make emphasis just on English, leaving this to editors.

Answer: Thanks for the reviewer’s comment. The English writing indeed could be further improved. We have asked for some help from some colleagues who are good at English writing. We have corrected the grammar errors. We wish the modification can satisfy the requirement.

  1. On lines 19-22 (and in the rest of the paper) authors mention pressure sensitivity of the acoustic vector sensor, or acceleration sensor, etc. Ideally, the velocity sensor should not respond to pressure, it should be sensitive only to a single parameter, so this influence is purely parasitical (in this particular case, it may be the hydrostatic sensitivity of the fibre laser). But in the paper, authors call acceleration response or velocity response a “pressure sensitivity” for unknown reasons.

Answer: For underwater acoustic field, velocity , displacement , acceleration , and pressure gradient  have relationship with pressure p as following:

Where  is unit vector in the sound propagation direction,  is the sound angular frequency,  is the sound speed,  is the water density. From the above equations, we know that the vector parameters are all related with the sound pressure. Once we get one of the vector parameters, the other three can be obtained through the above equations. The velocity sensor responds to the sound velocity. The amplitude of the sound velocity is determined by sound pressure. So it cannot be said the velocity sensor does not respond to pressure. Similarly, the acceleration sensors respond to sound acceleration. The amplitude of sound acceleration is also determined by sound pressure.

In the field of underwater sound, the response of the vector sensor to the sound is usually measured by the pressure sensitivity. The pressure sensitivity of the acceleration sensor is linearly increasing with the frequency. The pressure sensitivity of the displacement sensor is inverse proportional with the frequency. The pressure sensitivity of the velocity sensor keeps constant with the frequency.  When the frequency response of the pressure sensitivity of a vector is measured, the kind of the vector sensor is clearly understood.

  1. Also, throughout the entire paper, and on all their graphs, they call sensor “sensitivity” what is in fact the sensor “response”. The sensitivity is the ability to sense smallest signals, and in the current case is limited to the laser noise. The graphs (all of them) also do not show units on vertical axis.

Answer: Yes, it is true that there is difference on the definition of ‘sensitivity’ between the fields of underwater sound and photoelectric inspection. The sensitivity in the field of photoelectric inspection is the ability to sense smallest signals, and in the current case is limited to the laser noise. But in the field of underwater sound, sensitivity reflects the ability of the sensor response to the undertest parameter, as shown in section 2.4 of the manuscript. We have to explain that the definition of sensitivity in underwater sound field is not fabricated by us. Maybe because the research on water sound is not as popular as that on photoelectric inspection, the definition of sensitivity of the vector sensor is a little hard to accept for the researchers not in this field.

  1. The section 2.2 (theoretical model) seems to have very little connection with the rest of the text. For example, ln 157-160 authors claim that the shape of W(x/w) can be obtained but they do not present this, even for comparison with a well-known from mechanics formula of bent cantilever beam.

Answer: The shape of W(x/w) of the cantilever cannot be directly measured. The shape of the cantilever affects the shape of the DFB fiber laser. The strain of the DFB fiber laser is determined by the shape. The frequency variation of the DFB fiber laser is determined by the strain, i.e. the shape of W(x/w) of the cantilever. The calculation on the shape of W(x/w) is the basement for the simulation of the sensor sensitivity. The sensor sensitivity is the main point of the manuscript. And considering what can be measured is the sensitivity, so we did not present the results on the simulation of the shape of W(x/w).

  1. In section 2.3 (ln. 171-174) authors claim that formulas (14) (15) can be obtained from section 2.2. I believe it is wrong, and there is no such connection. Sa and Sp simply would follow from authors’ measurement setup (which was not though explained in enough detail)

Answer:  and  presented the relation between the phase variation of the interferometer and the sound acceleration and sound pressure. The phase variation of the interferometer has a linear relationship with the frequency variation of the fiber laser, which is shown in the section 2.1. Sa and  have a linear relationship with the frequency variation of the fiber laser. Section 2.3 presented the relation of frequency variation with the strain of the fiber laser (i.e. the shape of the cantilever). So Sa and  have relations with section 2.3. But we find that the sentence of ‘From the above analysis, the formulas of the acceleration sensitivity  and the acoustic pressure sensitivity  can be obtained as follows’ is not accurate. It is true that the formula of the sensitivity is not from the above analysis. We have changed this sentence to ‘the expressions of the acceleration sensitivity  and the acoustic pressure sensitivity  are as following for the sensor’.  

  1. See ln.175 g- is not a “gravitational constant” (and never was), it is a free-fall acceleration near the Earth surface. And certainly, has nothing to do with the subject of this paper. I do not know why authors have chosen 9.8m/sec2 as a reference and not just 1m/sec2.

Answer: Thanks very much for the reviewer’s comment. Yes, it should be called gravitational acceleration. It is a constant, but it is not called gravitational constant. It is our fault not to find this error. We have corrected it.

As to why we chosen 9.8m/sec2 as a reference and not just 1m/sec2, we have to say it usually chose one gravitational acceleration as the reference in our field. Of course, 1m/sec2 also can be chosen. But 9.8m/sec2 is more popular.

  1. There are serious flaws with the experiment (ln 295 onwards). The authors used a standard hydrophone as a reference for a vibration sensor, which is totally crazy, as they respond to different parameters. Moreover, the “standard hydrophone” is placed within 25cm of the water surface (pressure release surface) where (in theory), the acoustic pressure is zero. In current configuration, the bottom and the sides of their “standing wave tube) are the sources of secondary acoustic waves (in the sensors’ proximity) and it is not easy to predict the measurement result. In my view this invalidates any data obtained in the measurement.

Answer: The experimental setup is a standard instrument for calibrating the performance of vector hydrophones. The amplitude of sound velocity is related to sound pressure. The standard hydrophone is to measure sound pressure.

There is a shaker on the bottom of the standing wave tube. The height from the water surface to the tube bottom is L. The shaker generates vibrations (i.e. sound) in the water. Because of the size of the tube (tube diameter is larger than the height of the water in the tube), the sound in the tube is a standing wave. The pressure under the water surface is , where k is wave number, A and B is coefficients determined by the tube, x is the distance from the water surface. When ,. When , . So the pressure at the depth of d is . The pressure at each depth of the water in the tube is a certain value. So it is easy to predict the measurement. The data obtained in the measurement are validated.

  1. Statements in Ln 339-344 are plain wrong. You can’t enhance the sensor response (let alone the sensitivity) just by manipulating your demodulation system.

Answer: The frequency variation of the sensor has nothing to do with the demodulation system. The phase variation is dependent on the optical path difference of the interferometer. What we finally considered is the phase variation. The interferometer is part of the sensing system. It works as an amplifier to some extent. From the view of phase sensitivity of the sensor, it can be enhanced by optimizing the OPD.   

  1. Discussion on the fiber laser noise (ln345-347) seem to be totally out of context. But it is what actually defines the real “sensitivity” of the fiber laser sensor. I also totally do not understand why authors are always using “rad” units instead of “Hz” units throughout the entire paper. The phase is what is obtained by passing the fiber path imbalance, so why the sensor itself and its response (Hz/Pa, or Hz/(m/sec2)) is assumed to have connection with authors’ particular demodulation system (and the length of the path imbalance) is unclear.

Answer: Fiber laser noise determines the minimum signal that can be detected by the sensor. It is related to an important performance of ‘equivalent noise sound pressure’. As to why the ‘rad’ is used as the unit, that is because we take the interferometer as part of the sensor.  

Round 2

Reviewer 3 Report

I have read and considered the authors' response to my original comments. Unfortunately, I am not satisfied with their arguments. The authors decided to fend off most of the critical (in my view) issues with their paper rather than make a genuine attempt to make improvements.

In my view, authors' arguments are not convincing at, moreover, they are disappointing.  I no longer believe that the paper can be improved in its current form and have to reject it.

I hope that the authors would be able to resubmit a much better paper on the subject in the future.

Author Response

1. There are some flaws with English. (lines 29-33, 56-57 or 274-275 etc.). (Of course, a sphere filled with castor oil and fulfilled with castor oil is not the same thing. But I will not make emphasis just on English, leaving this to editors.
Answer: Thanks for the reviewer’s comment. The English writing indeed could be further improved. We have asked for some help from some colleagues who are good at English writing. We have revised the English expression and the grammar errors throughout the entire article. We wish the modification can satisfy the requirement.
2. On lines 19-22 (and in the rest of the paper) authors mention pressure sensitivity of the acoustic vector sensor, or acceleration sensor, etc. Ideally, the velocity sensor should not respond to pressure, it should be sensitive only to a single parameter, so this influence is purely parasitical (in this particular case, it may be the hydrostatic sensitivity of the fibre laser). But in the paper, authors call acceleration response or velocity response a “pressure sensitivity” for unknown reasons.
Answer: Thanks very much for the reviewer’s comment. We shall explain the question. The reviewer doubted that “Ideally, the velocity sensor should not respond to pressure, it should be sensitive only to a single parameter”. Yes, the velocity sensor should respond to the velocity, not the pressure. But the amplitude of the velocity is related with the pressure. Not only the amplitude of the velocity, but also the amplitude of the acceleration, displacement, and pressure gradient are all related with the pressure. The relationship between underwater acoustic velocity u ⃑, displacement D ⃑, acceleration a ⃑, pressure gradient ∇ ⃑p and the pressure p are as following:

Where n ⃑ is unit vector in the sound propagation direction, ωis the sound angular frequency, c_0 is the sound speed, ρ_0is the water density. 
Secondly, the particle acceleration or the particle velocity cannot be directly measured in the water since there are no relative standard measuring instruments. In air, as in Figure 6, a shaker can generate acceleration on the sensor and the magnitude of the acceleration can be measured by a standard accelerometer in a shaker. But when the sensor is in water, there have been no standard instruments to directly measure the magnitude of the water acceleration or velocity. In underwater sound field, only the pressure can be directly measured and the measuring instrument is named the standard hydrophone, as shown in Figure 11, which is a piezoelectric hydrophone. We use the standard hydrophone to measure the pressure of the vector sensor. Then the magnitude of the velocity or the acceleration, or pressure gradient can be obtained through the above equations. Finally, the response of the sensor to the velocity can be obtained. 
Thirdly, there is a specification that stipulates how to measure the sensitivity of a vector sensor. We followed the trade standard. Some other reports also followed the same standard, as in Journal of Lightwave Technology, 2012, 30(8): 1178 (DOI: 10.1109/JLT.2011.2170959), Applied Optics, 2018, 57(30): 9195 (DOI: 10.1364/AO.57.009195). The pressure sensitivity of an acoustic vector sensor has been reported in many other papers. For example, the phase sensitivity (dB re rad/μPa) of an accelerometer was presented in Journal of Lightwave Technology, 2012, 30(8): 1178 (DOI: 10.1109/JLT.2011.2170959).
In all, firstly, the magnitude of the velocity (also the acceleration, the displacement, the pressure gradient) is related with the pressure; secondly, only the water pressure can be directly measured by a standard instrument, so the pressure sensitivity of the acoustic vector sensor is mentioned; thirdly, the sensitivity of an acoustic vector sensor are measured following a set of trade standard. 
Finally, we think the reviewer gave us a very good suggestion that the usage of “pressure sensitivity” may bring some misunderstanding for the readers. So we followed the wording in other papers that we use “phase sensitivity” instead of “pressure sensitivity”. We have changed the “pressure sensitivity” in the manuscript to “phase sensitivity”. 
3. Also, throughout the entire paper, and on all their graphs, they call sensor “sensitivity” what is in fact the sensor “response”. The sensitivity is the ability to sense smallest signals, and in the current case is limited to the laser noise. The graphs (all of them) also do not show units on vertical axis.
Answer: Yes, there is difference on the definition of “sensitivity” between the fields of underwater sound and photoelectric inspection. The sensitivity in the field of photoelectric inspection is the ability to sense smallest signals, and in the current case is limited to the laser noise. But in the field of underwater sound, sensitivity reflects the ability of the sensor response to the undertest parameter, as shown in section 2.4 of the manuscript. It is the definition in the acoustic water sensing. Many other published papers about acoustic hydrophones all used this definition about sensitivity (DOI: 10.1109/JLT.2011.2170959; 10.1364/AO.57.009195; 10.1109/COA50123.2021. 9520010). 
As to the question that “The graphs (all of them) also do not show units on vertical axis”, we have added the unit with “dB re rad/μPa” to make it clear.
4. The section 2.2 (theoretical model) seems to have very little connection with the rest of the text. For example, ln 157-160 authors claim that the shape of W(x/w) can be obtained but they do not present this, even for comparison with a well-known from mechanics formula of bent cantilever beam.
Answer: The shape of W(x/w) of the cantilever cannot be directly measured. The shape of the cantilever affects the shape of the DFB fiber laser. The strain of the DFB fiber laser is determined by the shape. The frequency variation of the DFB fiber laser is determined by the strain, i.e., the shape of W(x/w) of the cantilever. The calculation on the shape of W(x/w) is the basement for the simulation of the sensor sensitivity. The sensor sensitivity is the main point of the manuscript. And considering what can be measured is the sensitivity, so we did not present the results on the simulation of the shape of W(x/w). 
5. In section 2.3 (ln. 171-174) authors claim that formulas (14) (15) can be obtained from section 2.2. I believe it is wrong, and there is no such connection. Sa and Sp simply would follow from authors’ measurement setup (which was not though explained in enough detail)
Answer: S_a and S_p presented the relation between the phase variation of the interferometer and the sound acceleration and sound pressure. The / variation of the interferometer has a linear relationship with the frequency variation of the fiber laser, which is shown in the section 2.1. S_a and S_p have a linear relationship with the frequency variation of the fiber laser. Section 2.3 presented the relation of frequency variation with the strain of the fiber laser (i.e., the shape of the cantilever). So S_a and S_p have relations with section 2.3. But we find that the sentence of ‘From the above analysis, the formulas of the acceleration sensitivity S_a and the phase sensitivity S_p can be obtained as follows’ is not accurate.  It is true that the formula of the sensitivity is not from the above analysis. We have changed this sentence to “the expressions of the acceleration sensitivity S_a and the phase sensitivity S_p are as following for the sensors:” on lines 189-191.
6. See ln.175 g- is not a “gravitational constant” (and never was), it is a free-fall acceleration near the Earth surface. And certainly, has nothing to do with the subject of this paper. I do not know why authors have chosen 9.8m/sec2 as a reference and not just 1m/sec2.
Answer: Thanks very much for the reviewer’s comment. Yes, it should be called gravitational acceleration. It is a constant, but it is not called gravitational constant. It is our fault not to find this error. We have corrected it. 
As to why we chosen 9.8m/sec2 as a reference and not just 1m/sec2, we have to say it usually chose one gravitational acceleration as the reference in our field. Of course, 1m/sec2 also can be chosen. But 9.8m/sec2 is more popular.
7. There are serious flaws with the experiment (ln 295 onwards). The authors used a standard hydrophone as a reference for a vibration sensor, which is totally crazy, as they respond to different parameters. Moreover, the “standard hydrophone” is placed within 25cm of the water surface (pressure release surface) where (in theory), the acoustic pressure is zero. In current configuration, the bottom and the sides of their “standing wave tube) are the sources of secondary acoustic waves (in the sensors’ proximity) and it is not easy to predict the measurement result. In my view this invalidates any data obtained in the measurement.
Answer: The experimental setup is a standard instrument for calibrating the performance of vector hydrophones, as shown in Journal of Lightwave Technology, 2012, 30(8): 1178 (DOI: 10.1109/JLT.2011.2170959), Applied Optics, 2018, 57(30): 9195 (DOI: 10.1364/AO.57.009195). The amplitude of the particle velocity cannot be directly measured in the water since there are no relative standard measuring instruments. But the amplitude of sound velocity is related to sound pressure. The standard hydrophone is to measure sound pressure. Through the measured pressure, the velocity can be derived. This is why we “authors used a standard hydrophone as a reference for a vibration sensor”. 
As to the question that “In current configuration, the bottom and the sides of their “standing wave tube) are the sources of secondary acoustic waves (in the sensors’ proximity) and it is not easy to predict the measurement result”, we have to firstly explain that the current configuration in Figure 11 is a set of standard measurement to calibrate a vector hydrophone, as in Journal of Lightwave Technology, 2012, 30(8): 1178 (DOI: 10.1109/JLT.2011.2170959), Applied Optics, 2018, 57(30): 9195 (DOI: 10.1364/AO.57.009195), and many other papers on acoustic vector sensors. 
Now I shall explain that the water sound filed in the standing wave tube is certain and predictable. There is a shaker on the bottom of the standing wave tube. The height from the water surface to the tube bottom is L. The shaker generates vibrations (i.e., sound) in the water. Because of the size of the tube (tube diameter is larger than the height of the water in the tube), the sound in the tube is a standing wave. The pressure under the water surface is P(x)= , where k is wave number, A and B is coefficients determined by the tube, x is the distance from the water surface. When x=0, p(0)=0. When x=L, P(L)= . So the pressure at the depth of d is P_d=(P_L sin kd)/(sin kL) e^jωt. The pressure at each depth of the water in the tube is a certain value. The vector sensor and the standard hydrophone are at the same depth below the water surface, so they experience the same acoustic pressure. The data obtained in the measurement are validated.
8. Statements in Ln 339-344 are plain wrong. You can’t enhance the sensor response (let alone the sensitivity) just by manipulating your demodulation system.
Answer: The frequency variation of the sensor has nothing to do with the demodulation system. The phase variation is dependent on the optical path difference of the interferometer. What we finally considered is the phase variation. The interferometer is part of the sensing system. It works as an amplifier to some extent. From the view of phase sensitivity of the sensor, it can be enhanced by optimizing the OPD.
We have to say that the phase sensitivity can be improved by enlarging the OPD, but the phase noise is also increased accordingly. The interferometer amplifies the amplitude of the fiber laser frequency variation. It also amplifies the phase noise level of the sensor. So the SNR is not improved. We thank the reviewer for the comment. We think it is not all-round to write the benefit of phase sensitivity improvement without discuss the disadvantage of phase noise deterioration. So we delete the statements in Ln 357-370.
9. Discussion on the fiber laser noise (ln345-347) seem to be totally out of context. But it is what actually defines the real “sensitivity” of the fiber laser sensor. I also totally do not understand why authors are always using “rad” units instead of “Hz” units throughout the entire paper. The phase is what is obtained by passing the fiber path imbalance, so why the sensor itself and its response (Hz/Pa, or Hz/(m/sec2)) is assumed to have connection with authors’ particular demodulation system (and the length of the path imbalance) is unclear.
Answer: Fiber laser noise determines the minimum signal that can be detected by the sensor. It is related to an important performance of ‘equivalent noise sound pressure’. As to why the ‘rad’ is used as the unit, that is because the frequency of the fiber laser cannot be directly measured. We transform the frequency to the phase signal of the interferometer. We also take the interferometer as part of the sensor. And we took the parameter of phase to judge the performance of the sensor. We plan to integrate the interferometer into the sensor sphere.
